# Influence of the Washing Process and the Time of Fruit Harvesting Throughout the Day on Quality and Chemosensory Profile of Organic Extra Virgin Olive Oils

**DOI:** 10.3390/foods11193004

**Published:** 2022-09-27

**Authors:** M. Pilar Segura-Borrego, Rocío Ríos-Reina, Antonio J. Puentes-Campos, Juan G. Puentes-Campos, Brígida Jiménez-Herrera, Pedro Vallesquino-Laguna, Raquel M. Callejón

**Affiliations:** 1Área de Nutrición y Bromatología, Department de Nutrición y Bromatología, Toxicología y Medicina Legal, Facultad de Farmacia, Universidad de Sevilla, C/P. García González n°2, 41012 Seville, Spain; 2Servicio Promoción Rural, Delegación Territorial de Agricultura, Pesca, Agua y Desarrollo Rural en Granada, Avenida Joaquina Eguaras 2, 18013 Granada, Spain; 3University Institute of Research on Olive Groves and Olive Oils, GEOLIT Science and Technology Park, University of Jaen, 23620 Mengibar, Spain; 4Department of Technology, Postharvest, and Food Industries, IFAPA “Cabra” Centre, Institute for Research and Training in Agriculture and Fisheries, Ctra. Cabra-Doña Mencía, KM. 2.5, Cabra, 14940 Cordoba, Spain; 5Departamento de Bromatología y Tecnología de los Alimentos, ETSIAM, Universidad de Córdoba, Campus de Rabanales, 14071 Córdoba, Spain

**Keywords:** extra virgin olive oil, washing process, organic cultivation, harvest time through the day, *Picual variety*

## Abstract

In recent years, there has been a growing demand for organic extra virgin olive oils (OEVOOs) as quality products with greater added value. The aim of the present work was to determine whether the washing process and time of harvesting (morning or afternoon) plays an important role in the quality of Picual OEVOOs by studying quality parameters (degree of acidity, peroxide value, K_232_, K_270_, oxidative stability), and volatile and sensory profiles. With this in mind, olive fruits were harvested from an early stage of maturity (over a period of 32 days, during morning and afternoon hours), and then were analyzed considering two main blocks (washed and non-washed), each of which had two sample types (morning and afternoon). Among the analyses performed, it is noteworthy that the volatile profile was obtained by headspace solid-phase microextraction (HS-SPME) coupled to gas chromatography–mass spectrometry (GC-MS). Regarding the physicochemical quality parameters, as well as the sensory and volatile profiles, it is remarkable that there were no differences between the oils produced under the two treatments applied (washed/non-washed). However, the time of harvesting (morning or afternoon) did influence the volatile and sensory profile, with higher values in the oils obtained from fruits harvested in the morning, being statistically significant for the families of aldehydes, hydrocarbures, and lactones. Besides, the olives harvested during the mornings gave rise to oils with slightly higher values in the green and apple fruit attributes.

## 1. Introduction

Olive oil production, 98% globally, is mainly concentrated in the Mediterranean countries, where 99% of olive groves are located. The European Union (EU) is the world’s leading producer of olive oil with a production greater than 80% globally. Within the European producing countries, Spain is the leader in both grove area and production [1], representing approximately 64% of the EU and 40% of the world’s production according to data from the Olive Oil Council International (COI); more than 80% of the national production corresponds to Andalusia. In addition, Spain is the world’s leading exporter of olive oil.

In recent years, there has been a notable development of organic farming worldwide [2], and the cultivation of the olive tree is not an exception to this fact. Spain, and especially Andalusia, is one of the main productive locations of the organic sector from an economic point of view [1].

Organic agriculture, also called biological or ecological, can be defined as a production system that integrates agronomic, economic, ecological, and social aspects. It promotes and enhances the use of natural agricultural resources, the maintenance of plant and animal diversity, as well as soil fertility and health, and the conservation of fauna and flora, minimizing the environmental impact [2,3,4]. Organic farming is regulated in the European Union by the most recent Regulation (UE) No. 2018/848 [5], where production, labelling, and control standards are detailed. According to this legislation, the use of synthetic fertilizers, pesticides and insecticides is severely restricted. This type of agriculture aims to establish a sustainable management system for its activity, as well as to promote the production of high-quality products that respond to consumer demand for healthy products, produced by methods and processes that are not harmful to the human beings, animals, and plants, nor harmful to the environment and well-being in general [6], in other words, towards foods with a greater added value.

The regulation and evolution of the quality of olive oil in Spain is largely conditioned by the International Olive Council and by the European Union [7,8,9,10,11,12]. In this way, the definition of regulated quality is included in the Commission Regulation (EEC) No. 2568/91 [11] modified by the Commission Regulation (EC) No. 656/95 of 28 March 1995 [12]. The regulated quality is thus defined, giving rise to the classification of olive oils in different categories according to a series of parameters. When in a virgin olive oil one of the physical-chemical or sensory parameters does not meet the standards, it goes to the next category.

On the other hand, the aroma of food, has been an object of study for its characterization since it is considered one of the most relevant quality criteria [13]. However, in the case of organic food, and therefore organic olive oil, it has been scarcely studied.

As a consequence of the consumer’s greater demand for quality products, both the national and international markets are increasingly demanding in terms of quality. That is why, in recent years, the olive sector has opted for a growing technological modernization in order to improve product quality levels, making quality and certification a necessary element of differentiation. Thus, many producers and researchers have been dedicating great efforts to obtain and know how to produce high quality olive oils under a variable background of environmental, agronomic, or technological factors that are not always controllable [14,15,16,17,18,19,20,21,22]. However, the relationship between the final quality of olive oil, whether organic or conventional, and the preliminary operations carried out on the fruit, among which is the washing process, has hardly been studied [23].

The washing operation is carried out after the fruit is received at the plant; it takes place before other operations such as grinding, spinning or centrifugation are performed, and it is the producer who decides whether it is necessary or not. Normally, the washing is applied when the fruit is collected from the ground, as it may contain traces of dirt, small stones, mud, etc. It is normally used to clean the fruit from various contaminations, i.e., physical, chemical, and even microbiological. To carry out the washing procedure, the incoming raw materials are usually introduced into a washing machine designed for this purpose, where the impurities that may be attached to the fruits are eliminated.

The washing operation seems to be a simple task within the general olive oil production process, but nowadays it is not free of some controversy when deciding whether or not it is convenient to carry it out on the fruit, due to the belief that it might affect both the production process and the quality of the product. Several authors agree that the fruits that come from the ground must be washed to eliminate the aforementioned impurities and thus avoid quality losses [23,24,25,26,27]. At the same time, these same researchers showed in their work that when the fruits are picked directly from the tree, they do not need to be washed [24,25], since it can influence their polyphenol content. This is supported by the fact that water may adhere to the skin of the fruit and, subsequently, generate emulsions that make the olive oil extraction process difficult. Regarding oxidative stability and sensory quality, it is also believed that washing water can extract/dilute different polyphenols from the fruit skin that give the oil its stability and different sensory characteristics [23].

On the other hand, since organic olive oils are highly valued by the consumer for being quality products with greater added value, the main concern of organic oil producers is to know the best production conditions to obtain the best oils. Although it has been extensively studied how the maturity of the fruit influences the aroma and quality of the oils in order to determine the optimal harvesting time, it has not yet been studied whether harvesting at different times through the day may also affect or influence its quality.

In accordance with the above, it could be interesting to undertake an analysis that would delve into the influence of the washing process in the production of organic olive oil, as well as to study if the moment of harvesting influences the quality of OEVOOs. For this reason, and given the real need of producers to know whether these factors could have an important influence on the quality of the final OEVOO, research was initiated on this matter by the authors [28,29], showing initially that the washing operation alone did not significantly influence diverse physicochemical and sensory parameters of fruits harvested directly from the tree. Continuing with that research, the aim of this work is to determine if the washing operation, together with the harvesting moment, could both play an important role in a characteristic set of physicochemical, sensory, and volatile parameters of OEVOO. Taking into account the variety of analytes considered here, the results of this study could serve as a fundamental reference in this field for decision-making related to the harvesting and initial operations applied to olive fruits.

## 2. Materials and Methods

### 2.1. Chemicals

High-purity (>95%) analytical standards of the volatile compounds were obtained from commercial sources (Sigma-Aldrich, Madrid, Spain; Merck, Darmstadt, Germany; Fluka, Madrid, Spain). Water from a Milli-Q purification system (Millipore, Burlington, MA, USA) was used. A mixture of C_10_ to C_40_ straight-chain n-alkanes (50 mg L^−1^ in n-hexane) from Fluka, (Madrid, Spain), were used to determine linear retention index (LRI). Solvents used were of analytical or HPLC grade (Sigma-Aldrich).

### 2.2. Olive Fruit Samples

Olive fruits were collected from Picual olive trees cultivated in a certified organic dry-land farm since 2003 (by the Council Regulation (EC) No. 1804/99 [30] and Commission Regulation (EC) No. 889/2008 [31]), under organic crop production, located in Alcaudete (Jaén, Spain), which has a continental Mediterranean climate with dry summers and mild winters. As was described in [28,29], the plantation framework followed a traditional pattern (about 12 m × 12 m), and the fruits were only tree-harvested by using a vibration machine. The olive fruits were harvested very early, over a period of 32 days between October and November 2013, with an average ripening index (RI) close to 2.1 according to the method proposed by Ferreira et al. [32]. Samples were randomly taken in the field, and two batches of fruits (of around 15 kg each) were collected on the sampling days selected for this study (1, 3, 8, 15, 25, 29, and 32). Olives were not taken in any case from the ground since the intention was to obtain high-quality oils. In addition, the collection was carried out in the morning (from 8:00 to 12:00 am) and in the afternoon (from 13:00 to 15:00 pm) with the aim of studying how the harvesting moment through the day might affect the fruits. The batches of olives collected were later homogenized in the laboratory and split into two smaller samples (of 7.50 kg each) in order to have enough material to perform further analysis. During this stage, the characterization of the fruit was carried out by measuring the average weight of one hundred fruits and the RI.

### 2.3. Olive Washing

From the samples collected in the morning, which were available each day of sampling, one of them was washed and the other one was left intact. The same was performed for the samples collected in the afternoon. The laboratory-scale washing process was undertaken using two 50 L drums, each containing 20 L of drinking water and the olives from a specific sample [28,29]. To simulate industrial washing conditions, compressed air from a regulation tank maintained at 600 kPa was injected into each drum for 2 min, promoting agitation of the olives in the water. The washing water was renewed every five days to approximately reflect the fluid conditions commonly present in industrial washing systems.

### 2.4. Olive Oil Extraction

Olive oil extraction was performed by using the Abencor system (Abengoa S.A., Sevilla, Spain) as was described in [28,29]. The oil produced from each sample was stored at 4 °C until analysis.

According to the experimental design, in this study, a total of 28 samples were processed, which were analyzed in duplicate. For the determination of the sensory and volatiles profile, a selection of samples was made, considering 3 stages of fruit maturity: an initial stage (day 1), an intermediate stage (day 15) and a final stage (day 32). Within each stage, these 4 types of samples were analyzed: washed-morning (WM), washed-afternoon (WA), non-washed-morning (NWM), and non-washed afternoon, (NWA).

### 2.5. Analytical Determinations

#### 2.5.1. Fruit Characterization

-Weight of 100 fruits: In each of the samples, the fruit weight was determined as the weight of 100 drupes randomly picked from aliquots of samples that had been previously homogenized.-Ripening index (RI): The method followed to determine the RI was that proposed by Ferreira et al. [32], which consists of classifying the fruit into eight classes according to the color of the skin and pulp.

#### 2.5.2. Analytical Indices

The indices free acidity, UV spectrophotometric indices (K_232_, K_270_), and peroxide values were evaluated, in triplicate for each sample, according to the official methods defined in the Commission Regulation (EEC) 2568/91 [11] and subsequent amendments of the Commission Regulation (EEC) 2568/91 [12,28,33].

#### 2.5.3. Oxidative Stability

Rancimat method [34] was followed for determining the oxidative stability by using a Rancimat apparatus (Metrohm, Herisau, Switzerland).

### 2.6. Sensory Analysis

The tasting panel of the Regulatory Council of the Protected Designation of Origin “Priego de Córdoba” carried out the organoleptic assessments of this study in accordance with the IOC standards. This panel was formed by a group of 8–12 tasters, previously selected and trained according to the techniques pre-established by the IOC/t.20/Doc n. 15/Rev.2 September 2007, and the European Regulation CE 640/2008 [33]. The sensory profile of each OEVOO sample was expressed as the median of each descriptor.

The olfactory sensations, such as green fruity, green leaf, fresh-cut grass, apple, almond, tomato, and other positive attributes, such as gustatory sensations (bitterness and sweetness) and tactile/kinaesthetic sensations (pungency), were directly or retronasally assessed by the panel. The intensity of the different descriptors considered in this study was rated into a continuous scale from 0 to 10 by each panelist.

### 2.7. Determination of the Volatile Profile

The determination of volatile compounds was carried out following the method proposed by Ríos-Reina et al. [35]. It consists of a headspace extraction using polydimethylsiloxane (PDMS) coated magnetic stir bars, called twister^®^, followed by an analysis by gas chromatography coupled to mass spectrometry (HSSE-GC-MS). The conditions were as follows: 2 g of sample were weighed in a 20 mL glass vial from Gerstel (Müllheim and der Ruhr, Germany). Then, after inserting the Twister^®^ in a glass insert, it was hermetically closed and heated at 60 °C for one hour in a thermostatic bath. The gas chromatograph (GC) was a 6890 Agilent system coupled with a quadrupole mass spectrometer (MS) Agilent 5975inert and equipped with a Gerstel Thermo Desorption System (TDS2) and a cryo-focusing CIS-4PTV injector (Gerstel).

Desorption was carried out in splitless mode, with a flow rate of 70 mL/min. The desorption program consisted of 35 °C during the first minute and a gradient of 60 °C/min until reaching 210 °C; these conditions were maintained for 5 min. The CIS-4 PTV injector was kept at −35 °C during the desorption process and then the temperature raised to 260 °C (10 °C/s) maintaining said temperature for 4 min.

The characteristics of the column used are the following: CPWax-57CB, 50 m × 0.26 mm, 0.20 µm thick (Varian, Middelburg, The Netherlands), and the carrier gas used is helium at a flow rate of 1 mL/min. The oven temperature program is 35 °C for 4 min, and an increase to 220 °C (2.5 °C/min), maintaining that temperature for 15 min.

The quadrupole, source and transfer line temperatures were maintained at 150 °C, 230 °C and 280 °C, respectively. The electron ionization mass spectra in the full-scan mode were recorded at 70 eV in an electron energy range between 29 and 300 *m/z*.

The NIST 98 library was used for the preliminary identification of the volatile compounds. Then, volatile compound identification was performed by comparing their linear retention index (LRI) with those of standards. When standards were not available, the compounds were identified by comparing the LRIs with the LRIs of standards described in the literature and by comparing the spectra with the NIST library as well as the LRIs with those of online databases (Flavornet; Pherobase) and the literature. LRIs were calculated by using the retention times of n-alkanes analyzed under the same analytical conditions. The samples were analyzed in triplicate.

### 2.8. Statistical Analysis

Data processing and volatile compound identification was carried out using the Deconvolution and Identification System software (PARADISe). Chromatographic data was converted into netCDF format and exported to AIA format using MSD ChemStation (version F.01.01.2317). After that, a total of 148 intervals were defined, and 8 compounds were extracted for each of them; this allowed the PARAFAC2 model to resolve the underlying and overlapping compounds in each interval [36]. For the correct selection of the final number of components, the fit and core-consistency was carefully optimized, trying to achieve values close to 100% for each parameter. The NIST MS Search Program (version 2.0) was used for preliminary identification of all the components of each model.

The significant differences at the 5% level between the means of the areas were determined by performing one-way ANOVA followed by a Tukey’s post hoc analysis, using INFOSTAT software (FCA, Universidad Nacional de Córdoba, Argentina).

A principal component analysis (PCA) was developed from the results of the physicochemical parameters and oxidative stability obtained, as well as on the results of the volatile composition, to observe if the washing of the fruits and the time of harvesting (morning or afternoon) and the day, influence the quality of the oils. It was undertaken using the PLS_Toolbox 7.9.5 (Eigenvector Research Inc., Wenatchee, WA, USA) working in a MATLAB environment. Data were autoscaled prior to PCA modelling.

## 3. Results and Discussion

### 3.1. Influence of Washing and Time of Harvesting on Agronomic and Physicochemical Parameters

Agronomic parameters such as weight of fruit and RI were determined to characterize the fruits (raw material). Then, physicochemical parameters such as free acidity, peroxide value, and UV spectrophotometric indices (K_232_, K_270_) and oxidative stability were determined as quality parameters to study if the fruit washing operation or even if the time of harvesting might have an important influence on the quality of the obtained OEVOOs (see Table 1).

In line with preliminary research initiated by the authors [28], Table 1 provides a concise overview of the values of the parameters aforementioned for a selection of representative days (1, 3, 8, 15, 25, 29, and 32), which collectively cover the sampling period.

The average weight of 100 fruits was approximately 304 g, with a standard deviation of around 32 g. No correlation or dependence was observed between weight and time over the evaluation period, suggesting that growth had ceased before harvest and that the olives experienced no significant moisture loss or gain during sampling.

According to that table, the RI showed a clear upward trend for morning and afternoon samples, with minimum and maximum values of the order of 0.49–0.71 and 4.04–4.08. Bearing in mind the recommendations of some authors [37,38], who place the optimal time to start harvesting when the RI is close to 3.50, it is understood in this case that the harvesting tasks were developed in very early stages of maturation. However, recent studies [39,40] showed that for olive groves of the Picual variety, very early harvesting tends to be associated with better quality and stability of the oils. Similarly, according to other researchers [37,38], the fat yield over dry matter in the Picual variety is typically around or above 40% when the RI exceeds 2. This supports the harvest calendar normally chosen by the owners of the farm that supplied the samples for this study, as was initially suggested by the authors in [28].

The degree of acidity (%) of the non-washed (NW) samples was slightly higher than that of the washed ones (W), and the difference was statistically significant (*p*-value < 0.05). However, this difference could be considered negligible, as all samples were well below the maximum legal limit of 0.80% established for extra virgin oils. This suggests that the difference between both treatments (washing vs. non-washing) is not relevant. In addition, no significant differences were observed between the samples from fruits collected in the morning and those collected in the afternoon, regardless of the treatment.

The peroxide index, expressed as active meq O_2_/kg oil, initially showed a slight upward trend that stabilized at the middle and final stages of the sampling period. This evolution differs from that reported by Jiménez et al. [19] and Salvador et al. [41], who observed an inverse proportional relationship between the RI and the peroxide index, which they attributed to the increase in lipoxygenase activity. However, our results are in concordance with other authors [42]. On the other hand, it is noteworthy that the washing process and harvesting time did not affect this analyte. No significant differences were observed between the washed and non-washed samples, regardless of harvesting time (*p*-value > 0.05 according to the data in Table 1 and the preliminary results from [28]). In any case, the values of all the analyzed samples were quite far from the maximum legal limit of 20 meq active O_2_/kg oil, a fact that reflects that the oxidation state of the analyzed oils was very low.

The absorbance values in the ultraviolet region of the electromagnetic spectrum (K_232_ and K_270_) also did not show significant differences between washed/non-washed or morning/afternoon samples (see Table 1); practically all of them were below the legal limits for OEVOO (2.5 for K_232_ and 0.22 for K_270_), except for three cases linked to K_232_ (day 1, non-washed/afternoon, and days 3 and 25, washed/afternoon), which can be considered anomalous measurements as detailed in [28]. Additionally, the decreasing trend observed in the K270 values for both washed and non-washed samples is notable, in accordance with Jiménez et al. [19], which could be related to the chlorophyll content [19,39]. As the authors preliminarily stated in [28], this fact agrees with the values obtained for the RI of the samples, whose mean value was around 2.10. The presence of these pigments (chlorophylls) has been related to possible unfavorable evolutions in oxidative activity in oils [42,43]. However, given the oxidative stability time shown in the Rancimat analysis, 6–10 days (144–240 h) in almost all the samples (Table 1), it seems that, in our case, the presence of chlorophylls did not have a relevant effect, probably due to the conditions of conservation of the samples (refrigeration at 4 °C and preserved from light).

The Rancimat results are noteworthy, since, although the fruits were collected in a state of early maturity, the organic oils of the Picual variety treated in this study, in addition to being of high quality, presented high values of oxidative stability, which is consistent with that reported by the authors in [28].

Considering the results presented in other works [40,44], the decreasing trend of oxidative stability time could be partially explained by the increase in the RI of the collected fruits. Regarding the washing process, unlike what happened with the degree of acidity, in the Rancimat tests, the non-washed samples presented a better performance than those that were washed, although these differences were not statistically significant (Table 1). Moreover, there is no marked difference in the values of this analyte between the morning and afternoon samples.

Hence, taking all the above into consideration, no relevant differences were observed, overall, in the physicochemical parameters just evaluated from samples treated by washing and non-washing processes. These results agree with those observed by Beltrán et al. [45], where no differences were found in the quality parameters between the washed and non-washed fruits stored in perforated boxes. However, Vichi et al. [46] observed lower values of the peroxide index and K_270_ in the oils obtained from washed and stored olives, possibly due to the anaerobic conditions that take place during storage. In our case, the olives were processed immediately after the washing process, so they did not have to be stored. Regarding the time of harvesting (morning or afternoon), no significant differences were observed in either any of the previously mentioned parameters, resulting, in all cases, in high quality oils (see also Figure 1a,b).

In line with the aforementioned, and with the aim of complementing and analyzing previous work from a different perspective, a principal component analysis (PCA) was performed on all the parameters addressed thus far. Figure 1a,b (just shown) displays the scores and loadings plots, respectively, obtained by the first and third principal components (PCs), which explain 51.61% of the variance. As could be seen, the physicochemical parameters, RI and oxidative stability basically allowed the samples to be differentiated according to the maturity stage of the fruit at the time of harvest. However, no differences were observed between the samples obtained with the washing and non-washing processes nor between those of fruits collected in the morning and in the afternoon, confirming the results discussed previously. Moreover, according to the loadings, initial samples seem to be more related to oxidative stability time and as expected, while late samples (harvested at 29 and 32 days) are more related to weight and RI.

### 3.2. Assessment of Sensory and Volatile Profile Results

#### 3.2.1. Sensory Profile

Consistent with Section 2.4, Table 2 provides a compact summary of the sensory attributes of the OEVOOs previously analyzed by the authors in [29]. In this case, these parameters were evaluated at three stages of fruit maturity (initial stage—day 1; intermediate stage—day 15; and final stage—day 32) for both washed and non-washed samples collected in the morning and afternoon. It can be noted from the table that green fruity (average value AV = 5.30) was the most prominent sensory attribute under all conditions (washing/non-washing and morning/afternoon harvesting times). Other descriptors such as bitterness (AV = 4.00), pungency (AV = 3.50), and green leaf (AV = 3.30) were also notable, though to a lesser extent. The remaining attributes (fresh-cut grass, sweet, almond, apple, and tomato) were present in the OEVOOs tasted, but with lower intensity, as can be verified in Table 2.

These results are in agreement with the literature, since oils of the *Picual variety* are described as oils with “a great personality”, full-bodied, and with a high green fruity score for the olives—appreciating the taste of the olive leaf, slight itching and bitterness, that intensifies when the fruit is very green [24]. As was previously stated in [29], it is also remarkable that no defects were detected in the analyzed samples, resulting in an extra virgin classification in all cases. According to various authors [19,39,41,47,48], this outcome is common in fruits with low ripening indices, as is the case of the present study (RI ranged from 0.49 to 4.08), which would support the sensory evaluation results obtained. Likewise, the filtration process applied during the production of the oils has also been able to help prevent the appearance of defects. According to Aguilera et al. [49], filtering removes impurities and vegetation water, which can result in greater stability of the oils over storage time.

For comparative purposes, the average attributes of the different types of samples considered in this study (WM, NWM, WA, NWA) are presented in a spider graph (Figure 2). Similarly to the results reported in [29], the sensory profiles of washed and non-washed samples were nearly identical, showing no significant differences in any of the sensory attributes analyzed with respect to the treatment applied to the fruits. However, regarding the harvesting time, some differences have been observed in the sensory profile of OEVOOs from fruits collected in the morning and in the afternoon. It is noteworthy that oils obtained in the mornings presented slightly higher values for the green fruity and apple attributes (see those marked with (*) in Figure 2b). These are positive attributes highly appreciated by consumers and, therefore, related to quality.

#### 3.2.2. Volatile Profile

In this study, a total of 58 volatile compounds were determined. Their total peak areas, as well as their standard deviation and statistical results (obtained by performing an ANOVA regarding the time of collection, maturation stage and washing and non-washing process), are shown in Appendix A. Among these 58 volatile compounds, 2 were acetic acid esters, 2 acids, 14 alcohols, 11 aldehydes, 7 hydrocarbures, 4 esters, 10 ketones, 4 lactones, 2 (sesqui)terpenes and 2 are classified as “others”.

As it is well known, the main volatile compounds in AOVs are the C_6_ and C_5_ derived from linoleic and linolenic acids through the lipoxygenase (LOX) pathway, which takes place during olive crushing and malaxation of fruit and olive paste [50]. These compounds, named LOX compounds, which are mainly alcohols, aldehydes, esters, hydrocarbures and ketones, are mainly responsible for the characteristic and positive green sensory notes.

In general terms, with respect to the values of the sum of the peak areas obtained for the total volatile compounds of the OEVOOs analyzed, no significant differences were observed with respect to the fruit washing process. However, some statistically significant differences were observed with respect to the time of harvest, both within the same day (in the morning or in the afternoon) and in the different stages of fruit maturity. Thus, OEVOOs obtained from fruits collected during the mornings generally presented higher values of total peak areas, being statistically significant for the families of aldehydes, hydrocarbures and lactones (Figure 3). This is in accordance with the sensory results, since the OEVOOs from olives collected in the morning generally reached higher values in the positive attributes, especially green fruity and apple, which seems to indicate that the harvest time gives rise to oils of greater aromatic complexity, and therefore of higher quality.

On the other hand, ketones, hydrocarbures and the family considered as other compounds presented the highest and statistically significant values of total areas in the initial stage of fruit maturity (day 1) than in the other two stages of the fruits (day 15 and day 32) collected during the mornings (Figure 3). In these OEVOOs there were also observed statistically lower values for the aldehyde family in the final stage (day 32). These results are in accordance with those described by other authors [51]. In addition, we can highlight the acetates and sesquiterpenes for reaching statistically higher values in the final stage and lower in the initial one of fruit maturity, especially in OEVOOs from fruits harvested in the afternoon. This was to be expected, since, according to the literature, sesquiterpenes have been described as markers of the state of maturity of the fruit [52].

To better study the differences in the volatile profile due to the washing and non-washing processes, harvest time (morning or afternoon) and fruit ripening stages (1, 15 and 32 days), a PCA with four PCs was carried out considering the volatile compounds as variables (Figure 4).

Figure 4a,b show scores and loadings plots, respectively, obtained by the first and second principal components (PC1 and PC2, respectively), which explain 56.07% of the variance. As can be seen in the scores plot (Figure 4a), the volatile composition essentially differentiated the samples according to the time of collection (morning or afternoon). Thus, the samples corresponding to fruits collected during afternoons (A) were placed on the upper and left side of the scores plot (negative values of PC1 and positive values of PC2) and the samples collected during the morning (M) on the lower and right side (positive values of PC1 and more negative values of PC2). There was also a certain grouping according to the day of harvesting. According to the loadings (Figure 4b), one can see again the relationship discussed above between hydrocarbons, aldehydes, and lactones with the morning samples (M), such that almost all the volatile compounds of these families are in the positive side of PC1, where morning samples were positioned, while esters such as isopropyl palmitate were more related to the afternoon samples (A). However, no differences were observed between the washed or non-washed samples, confirming the results discussed above.

In addition, to better see the differences according to the maturity stage of the fruit, PC3 and PC4 were represented (Figure 4c d), in which some differences could be observed throughout the fruit ripening process. Thus, when representing the scores and loadings obtained by the PC3 and PC4, it was possible to observe that the OEVOOs from the last stage of fruit maturity (32 days) were the most grouped, regardless of the washing process or the time of harvest (morning or afternoon), so they seem more similar with respect to the volatile profile. However, the OEVOOs of the initial and intermediate stages (1 and 15 days) turned out to be much more dispersed, which suggests that the conditions for obtaining OEVOOs have more influence in these stages; in this case, differences in volatile composition were, fundamentally, due to the time of collection (morning or afternoon). Regarding the loadings, hexyl acetate and *(Z)*-3-hexenyl acetate seem to be related to the samples from the final stage (32 days). These results are in agreement with the literature, since these two acetates are responsible of the fruity notes [53,54], and according to several authors [35,55]) Picual oils have shown to reach the maximum fruity sensory values in the last stage of fruit maturity. Moreover, aldehydes, such as *(E)*-2-hexenal or hexanal, seemed to be more related to intermediate and last stages, while alcohol volatile compounds, such as *(E)*-2-hexen-1-ol, and 1-hexanol, were more related to the first stages (day 1) according to the scores and loadings plot (Figure 4a,b), being again in agreement with the literature [35,55].

Within the chemical groups with the highest number of compounds, the alcohol family was the largest, followed by aldehydes and ketones. Alcohols have been described as the class with the greatest variety of volatile compounds in olive oils in qualitative terms [56]. In our study, ethanol was more related to OEVOOs from non-washed fruits, since it presented statistically higher peak areas in these samples (Appendix A). This result is contrary to that observed by Beltrán et al. [45], since in their study, washed fruit oils presented higher ethanol concentrations. This could be due to the fact that in that study, the fruits, once washed, were stored for a short period of time where ethanol biosynthesis might have been favored. Nevertheless, in our case, the fruits were processed immediately after washing. Therefore, the higher area values in the non-washed samples could be due to the fact that during the transport of the fruits, they can be subjected to high temperatures (and especially in an early harvest) and give off some humidity that favors the synthesis of ethanol in their surface, which could be washed away by the washing water. On the other hand, regarding the time of harvest, 2-hexyn-1-ol and *(Z)*-2-penten-1-ol showed to be more related to OEVOOs from olives harvested in the morning (greater peak areas), while the 2-hexen-1-ol was more related to the afternoon samples (Appendix A). Regarding the maturity stage of the fruit, significant differences were observed in the initial stage, reaching higher values of 1-dodecanol, 2-phenoxyethanol, 1-undecanol, 1-penten-3-ol and *(Z)*-2-penten-1-ol. On the contrary, in the final stage 1-hexanol presented statistically higher values, being in agreement with Ríos-Reina et al. [35].

Aldehydes seemed to be more present in OEVOOs obtained after harvesting during the morning. Thus, hexanal, 3-methyl-2-butenal, *(Z)*-2-hexenal and *(E)*-2-hexenal presented significantly higher values of total areas in the morning samples while only 2-undecanal presented higher values in OEVOO of in the afternoon (Appendix A). According to Angerosa et al. [57], hexanal and *(E)*-2-hexenal have proven to be two of the most important contributors to the high quality of OEVOO, which are responsible for green notes. This explains the differences observed in sensory profiles. On the other hand, nonanal, decanal, benzaldehyde and 5-methylfurfural were more related to the first stage of fruit maturity, while 2-hexenal, *(E)*-2-hexenal and *(E)*-2-nonenal presented higher peak area values in the last stage of maturity. This is in agreement with the literature since, according to some authors, 2-hexenal, responsible for fruity notes, contributes to the discrimination of the ripening stage [35,58].

Within the ketones, the third group with the highest number of compounds, they seemed to be more related to the morning harvest and with the first stage of maturity of the fruit. Thus, 1-penten-3-one, 1-hydroxy-2-propanone, 1-hydroxy-2-butanone, 1,2-cyclopentanedione and 2-hydroxy-3-methyl-2-cyclopenten-1-one presented significantly higher areas in the morning collected samples (Appendix A). These last two ketones, in addition to 2-pentanone and 2-octanone, reached the highest peak area values in OEVOO from the first stage of fruit maturity, which agrees with the literature [35]. Between the washed or non-washed samples, no significant differences were observed in any of the determined ketones. 2-Pentanone has been recognized as one of the main compounds responsible for fruity notes [59]; hence, this compound could be related to the marked fruity notes of the OEVOOs in the early stages.

Finally, the rest of the chemical classes, with a lower number of compounds than the other groups, could be considered a minority, as other authors also suggest [56]. Within them, hexanoic acid presented significant differences between the washed and non-washed samples, reaching the highest areas in the OEVOOs obtained after a washing process (Appendix A). This compound is related to fatty notes and fat oxidation, and is defined as a sensory defect, such as being rancid or vinegary [35,60]. However, none of the analyzed oils presented defects at the sensory level, probably due to the fact that the concentrations of this compound were below its olfactory detection threshold. In fact, this has been already identified in EVOOs at low concentration [61,62].

In summary, we could say that the time of harvest (morning or afternoon) had a greater influence on the volatile and sensory profiles compared to the washing and non-washing processes. The differences between both varieties were related to some alcohols and aldehydes C_5_ and C_6_, and with the sensory attributes fruity green and apple.

## 4. Conclusions

The regulated quality of olive oil depends on many factors throughout its production process, and in this work, it has been studied whether the washing of fruits can be one of them when it comes to making quality extra virgin olive oils from organic olive groves. That is why in this study the oils were obtained under conditions that allowed the highest quality OEVOO to be obtained, that is, in very early harvest conditions and only with flight olives. In addition, the time of harvest has also been considered, not only in different stages of maturity of the fruit (different days) but also in the same, that is, collecting them at different times of the day (in the morning and in the afternoon).

In general, according to the physicochemical quality parameters considered, it is relevant that, under the conditions in which this study was carried out (very early harvest and use only of flight olives), it cannot be affirmed, notably, that there are differences between the oils obtained from washed fruit and those that were made without prior to washing, nor between those obtained after harvesting during the morning and in the afternoon. Regarding the oxidative stability time, no significant differences were observed between washing and non-washing processes, nor between the time of harvest throughout the day, since all the OEVOOs analyzed showed high stability values. Regarding the sensory profile (green fruity, sweet, pungency, bitter, green leaf, fresh-cut grass, apple, almond and tomato) and volatile profile, it was clearly observed that there were also no differences between the oils produced under the two applied treatments (washing and non-washing processes).

However, the time of collection (morning or afternoon) did influence the volatile and sensory profile of OEVOO, generally presenting higher values of total peak areas in the oils obtained from fruits collected during the mornings, being statistically significant for the families of aldehydes, hydrocarbures, and lactones. Regarding the sensory profiles, the olives harvested during the mornings gave rise to oils with slightly higher values in the green and apple fruit attributes.

Being aware of the crisis scenario that surrounds us, and to which olive oil producers are not strangers, it is understood that the results of this work could be a consideration factor in order to make decisions about the most appropriate harvesting schedule, and on whether or not to wash the fruits, due to the cost savings that could be achieved without causing any loss of quality to organic oils.

Consequently, this study opens a line of research aimed at knowing the characteristics of organic virgin olive oils, as well as studying the conditions that allow the highest quality oils to be obtained.

On the other hand, the results of this work will be of great importance, due to its applicability, for the olive sector, since they will allow us to delve into the factors that allow an increase on the quality and performance of its oils, and especially of organic ones. All this will undoubtedly contribute to increase and consolidate the prestige of organic extra virgin olive oil, with the consequent impact on the strengthening of the foreign market.

## Figures and Tables

**Figure 1 foods-11-03004-f001:**
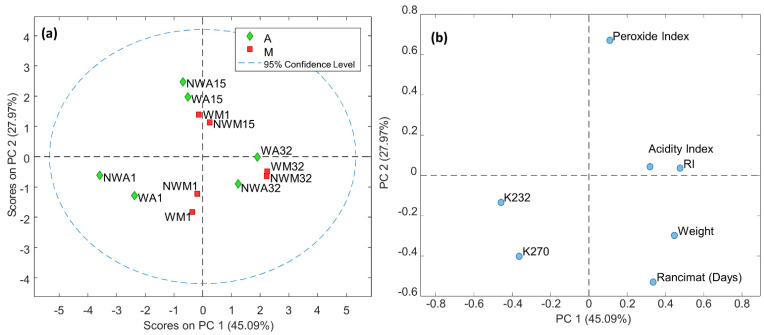
Scores (**a**) and loadings (**b**) plots of the PCA model obtained with physicochemical parameters, Ripening Index (RI), oxidative stability (Rancimat results) of the Organic Extra Virgin Olive Oil samples (OEVOOs) analyzed according to the washing (W) or non-washing (NW) process, the time of collection (Morning-M or Afternoon-A), and maturation stage (I–III). Stage I (collection day 1), stage II (collection day 15), stage III (collection day 32).

**Figure 2 foods-11-03004-f002:**
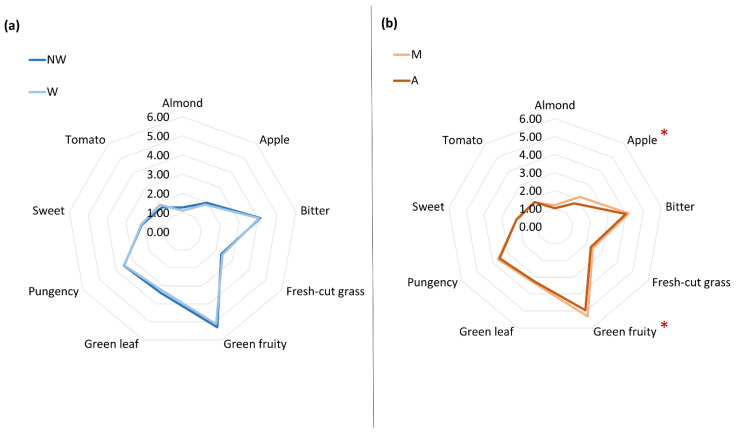
Comparison of the sensory profiles (odor attributes) between the Organic Extra Virgin Olive Oil (OEVOOs) obtained from washed (W) and non-washed fruits (NW) (**a**) and from fruits harvested during morning (M) or afternoon (A) (**b**).

**Figure 3 foods-11-03004-f003:**
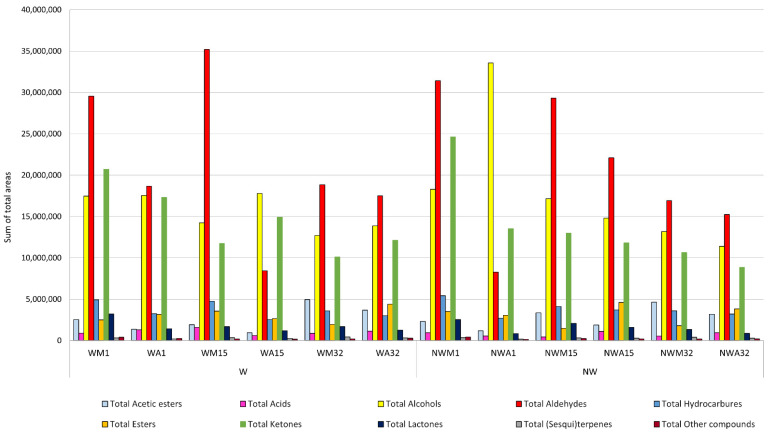
Areas of the different chemical families of volatile compounds determined in the different Organic Extra Virgin Olive Oil (OEVOOs) analyzed according to the washing (W) or non-washing (NW) process, the time of collection (Morning-M or Afternoon-A), and maturation stage (I–III). Stage I (collection day 1), stage II (collection day 15), stage III (collection day 32).

**Figure 4 foods-11-03004-f004:**
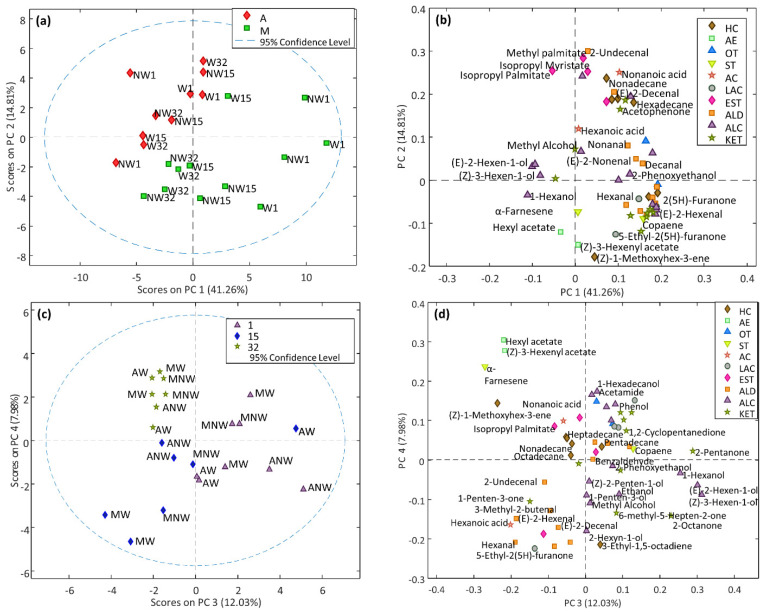
Scores (**a**–**c**) and loadings (**b**–**d**) plots of the PCA model of the first 4 principal components (PCs) obtained with the volatile compounds of the Organic Extra Virgin Olive Oil samples analyzed. PC1 vs. PC2 scores plot (**a**) was colored according to the time of collection, and PC3 vs. PC4 plot was colored according to the maturation stage. Notes: Morning (M) or afternoon (A), washing (W) or non-washing (NW) process; Stage I (collection day 1), stage II (collection day 15), stage III (collection day 32); HC: Hydrocarbures, AE: acetic esters, OT: other compounds, ST: (sesqui) terpenes, AC: acids, LAC: lactones, EST: esters, ALD: aldehydes, ALC: alcohols, KET: ketones.

**Table 1 foods-11-03004-t001:** Agronomic and physicochemical parameters of Picual Organic Extra Virgin Olive Oil (OEVOO) samples depending on the sampling day and the time of collection (morning or afternoon) and the washing process.

		Sampling Day	Average Weight (100 g)	Ripening Index (RI)	Acidity Index	Peroxide Index	K_232_	K_270_	Rancimat (Days)
**Non-washed (NW)**	**Morning (M)**	1	320	0.71	0.23	2.28	1.78	0.17	8.60
		3	291	1.1	0.29	2.46	2.25	0.18	8.32
		8	294	1.74	0.23	2.89	1.70	0.16	6.78
		15	318	1.82	0.22	2.85	1.59	0.16	6.13
		25	340	3.22	0.28	3.71	2.36	0.10	7.31
		29	318	4.08	0.29	2.46	1.66	0.12	7.55
		32	324	3.22	0.26	2.48	1.68	0.15	9.54
	**Afternoon (A)**	1	300	0.82	0.22	2.20	3.19	0.20	6.82
		3	302	0.49	0.27	4.51	1.74	0.20	8.11
		8	296	1.58	0.25	3.33	1.62	0.16	9.57
		15	298	1.45	0.26	3.31	1.83	0.16	5.93
		25	346	3.37	0.26	3.71	2.32	0.11	7.58
		29	298	4.04	0.27	2.47	1.78	0.12	8.24
		32	324	3.22	0.22	2.50	1.61	0.17	8.79
**Washed (W)**	**Morning (M)**	1	320	0.71	0.22	2.15	1.66	0.19	9.04
		3	291	1.1	0.20	2.91	2.16	0.23	8.31
		8	294	1.74	0.20	3.26	1.63	0.17	6.98
		15	318	1.82	0.17	2.85	1.81	0.13	6.49
		25	340	3.22	0.24	3.33	2.40	0.09	7.13
		29	318	4.08	0.22	2.49	1.64	0.13	7.13
		32	324	3.22	0.26	2.48	1.58	0.14	9.21
	**Afternoon (A)**	1	300	0.82	0.18	2.05	2.28	0.18	8.04
		3	302	0.49	0.23	2.88	2.74	0.18	7.50
		8	296	1.58	0.22	3.71	1.65	0.18	6.68
		15	298	1.45	0.22	2.87	1.64	0.14	6.39
		25	346	3.37	0.19	3.30	4.97	0.09	6.25
		29	298	4.04	0.20	2.47	1.81	0.13	5.96
		32	324	3.22	0.24	2.43	1.64	0.13	8.09

**Table 2 foods-11-03004-t002:** Sensory profile of Picual Organic Extra Virgin Olive Oil (OEVOO) depending on the sampling day (1, 15 and 32) and the time of collection (morning or afternoon) and the washing process.

		Day	Almond	Apple	Bitter	Fresh-Cut Grass	Green Fruity	Green Leaf	Pungency	Sweet	Tomato
**NW**	**M**	**1**	1.09	1.06	4.71	2.54	5.80	3.62	3.20	1.68	2.07
		**15**	2.43	2.77	4.77	2.99	5.51	3.06	3.6	2.6	0
		**32**	0	0	3.44	1.84	5.19	3.42	4.28	1.97	1.91
	**A**	**1**	0	1.41	4.4	2.3	5.61	3.65	3.25	2.11	1.96
		**15**	2.04	2.04	3.54	2.43	4.91	2.83	2.80	2.43	0
		**32**	1.4	0	3.49	2.35	5.87	3.46	4.44	1.99	1.76
**W**	**M**	**1**	1.79	2.11	4.18	2.36	5.77	3.2	3.84	2.04	2.22
		**15**	0	2.27	4.42	1.55	5.23	3.03	3.6	2.57	0
		**32**	2.21	0	3.33	2.47	5.19	2.83	4.57	2.28	2.57
	**A**	**1**	2	1.99	4.05	2.37	5.05	3.36	3.87	2.11	1.56
		**15**	1.14	0	3.02	2.13	4.21	1.99	2.17	2.89	2.94
		**32**	0	0	3.6	2.96	5.56	4.47	5.55	1.84	1.71

Note: Odor attributes assessed in a 1–10 scale. NW: Non-washed. W: Washed. M: Morning. A: Afternoon.

## Data Availability

Data will be made available on request.

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
