# Peer review of "Influence of the Washing Process and the Time of Fruit Harvesting Throughout the Day on Quality and Chemosensory Profile of Organic Extra Virgin Olive Oils"

_foods, 2022, doi:10.3390/foods11193004_

Round 1
Reviewer 1 Report
The manuscript “Influence of the Washing Process and the Time of Fruit Harvesting throughout the Day on Quality and Chemosensory Profile of Organic Extra Virgin Olive Oils” fits into the scope of the journal and is generally well-written. However, there is a number of aspects which should be addressed. Below I have provided specific comments & suggestions for the Authors:
Most serious issues:
The aim of washing is to clean the fruit from various (not only physical, as mentioned in the Introduction of the manuscript, but also various chemical and microbiological) contaminations, to ensure appropriate food safety standards. Thus, concluding about lack of necessity of fruit washing based only on the parameters measured in this study is in my opinion inappropriate. The text should thus be rewritten to make the above raised issue clear. This aspect should be consequently revised in all relevant sections of the manuscript (abstract, discussion, conclusions). Considering the above raised issue, the statements made in the manuscript about high applicability of this study for the producers are too strong.
Other comments & suggestions:
Abstract:
Line 20: I would suggest to delete “main”, since there are plenty other parameters that could be measured.
Introduction:
Line 44: A reference to information given in this line should be provided.
Lines 52-54: The 2 regulations mentioned here are not in force any more. Thus, the sentence needs revising.
Line 118-119: Language correction is necessary here.
Line 130: A full stop and space before the subsection title should be deleted.
Lines 130-146: Fonts in this subsection should be unified.
Line 132: Was this a certified organic farm? (following the EU regulation? As it was in 2019, previous EU regulation should be mentioned here).
Methodological sections:
Line 228: “(Ryan et al., 2004)” should be deleted.
Line 229: by triplicate -> in triplicate
Results:
Line 337: I suggest to delete “Determination of the” from the section title (as these are results and not methods).
Line 366: I suggest to delete “Composition of” from the section title (as the authors do not measure composition of compounds but their contents/profile).
As previously mentioned – tables and figures should be provided in the manuscript.
Additional comments for tables and figures :
Table 1: Title should reflect more clearly what’s in the table; I would avoid using text in capital letters in the rows’ headings; explanation of “TT” should be revised (ANOVA and Tukey’s test are two different things and statement “analysis of variance by tukey's test” is incorrect); explanation of the meaning of lowercase and capital letters seems to be incorrect and should be carefully revised (first of all it’s about difference between values within the same columns and not the same rows, and secondly there are e.g. no washing parameters in the table). Moreover, letters showing significance (or lack of significance) of differences between numbers should be placed next to every number and not in a separate row (what if there was a significant difference? How would this be indicated in this table?). Finally, why is Table 1 placed in the methodological section, and not in the Results section?
Figure 1, Figure 4, Figure 5: In the titles there is no information that it’s about olive oil. Figure title should be revised to reflect clearly what’s on the figure.
Table 2: Again – the title should reflect clearly what’s in the table (Sensory profile of Picual Organic Extra Virgin Olive Oil depending on…); in the footnote: A – Afternoon (and not “Afertnoon”); It should be specified in the table whether these are taste attributes (or odour); Statistical evaluation results are missing in the table; I would also suggest to specify (e.g. in the footnote) what the scale was for evaluation of these attributes.
Figure 2: Again, it should be specified if the attributes in the figure = taste attributes; Each figure should be self-explanatory, thus I would suggest not to use abbreviation in the title (OEVOOs), but full name; The meaning of stars shown on the figure should be explained in the footnote.
Figure 3: It should be clarified what the Authors mean by “total areas”; each Figure should be self-explanatory, thus I would suggest not to use abbreviation in the title (OEVOOs), but full name; Label should be added to y axis.
‘Technical’ aspects:
Authors’ names should not be written in capital letters.
Section and subsection titles – all words should begin with capital letters (except for “and”, “of” etc.).
Current section 3 (Analytical determinations) should be a part of section 2 (Materials and Methods), and the following subsections of current section 3 should become subsections of section 2. Consequently, “Results” would become section 3 (and should be renamed as “Results and Discussion”), and “Conclusions” would become section 4.
“Supplementary Materials” section: The following text should be provided and the following formatting should be applied in this section of the manuscript: “The following supporting information can be downloaded at: www.mdpi.com/xxx/s1, Table S1: title; Table S2: title.” Moreover, it would be good if the Authors could add a specific heading to the Supplementary file, to make it clear that this is a Supplementary file related to this manuscript.
Numbering of tables in the Supplementary Materials should be done with Arabic numerals (S1, S2), and this should be revised accordingly in the whole manuscript text.
“Author Contributions” section: initials (and not full names of the authors) should be given.
“Data Availability Statement’ section should be added (after “Funding” section).
“Institutional Review Board Statement” section: Please revise (now the instructions from the manuscript template are written in this section…).
“Conflicts of Interest” section: add full stop at the end of the sentence.
References:
Following the requirements of Foods (MDPI), all names of the journals in the listed references should be abbreviated. Other aspects should also be carefully revised, to ensure the style follows the requirements of the journal. At the moment there is a significant number of inconsistencies.
Author Response
Response to Reviewer 1 Comments
Point 1: The aim of washing is to clean the fruit from various (not only physical, as mentioned in the Introduction of the manuscript, but also various chemical and microbiological) contaminations, to ensure appropriate food safety standards. Thus, concluding about lack of necessity of fruit washing based only on the parameters measured in this study is in my opinion inappropriate. The text should thus be rewritten to make the above raised issue clear. This aspect should be consequently revised in all relevant sections of the manuscript (abstract, discussion, conclusions). Considering the above raised issue, the statements made in the manuscript about high applicability of this study for the producers are too strong.
Response: We agree with the reviewer’s advice. As mentioned in the Introduction, the washing process is usually applied when the fruit is collected from the ground, and it is normally used to clean the fruit from various contaminations. Although, in this work organic olives were not taken in any case from the ground to obtain high quality oils, we do not want to say that there is not never necessary to wash the fruit. We only could say that the washing did not affect the sensory and volatile profile of the organic olive oil obtained, so it is a decision of the producer to perform it according to the contaminants that he/she could know it could have. For this reason, we have made some modifications in the manuscript that clarify this aspect and make statements less hard than it seemed before.
ABSTRACT:
Point 2: Line 20: I would suggest to delete “main”, since there are plenty other parameters that could be measured.
Response: According to the reviewer, this change was made in the abstract.
INTRODUCTION:
Point 3: Line 44: A reference to information given in this line should be provided.
Response: According to the reviewer, a new reference was included in the revised manuscript.
Point 4: Lines 52-54: The 2 regulations mentioned here are not in force any more. Thus, the sentence needs revising.
Response: According to the reviewer, the sentence has been revised and modified. Now these 2 regulations are not mentioned in this sentence.
Point 5: Line 118-119: Language correction is necessary here.
Response: According to the reviewer, this change was made in the revised manuscript.
Point 6: Line 130: A full stop and space before the subsection title should be deleted.
Response: According to the reviewer, this change was made in the revised manuscript.
Point 7: Lines 130-146: Fonts in this subsection should be unified.
Response: This error was corrected in the revised version.
Point 8: Line 132: Was this a certified organic farm? (following the EU regulation? As it was in 2019, previous EU regulation should be mentioned here).
Response: It was a certified organic farm since 2003 according to the corresponding EU regulation. These details were included in the revised version of the manuscript.
METHODOLOGICAL SECTIONS:
Point 9: Line 228: “(Ryan et al., 2004)” should be deleted.
Response: This error was corrected in the revised version.
Point 10: Line 229: by triplicate -> in triplicate
Response: This error was corrected in the revised version.
RESULTS:
Point 11: Line 337: I suggest to delete “Determination of the” from the section title (as these are results and not methods).
Response: According to the reviewer, this title was rewritten as follows: Assessment of sensory and volatile profile results.
Point 12: Line 366: I suggest to delete “Composition of” from the section title (as the authors do not measure composition of compounds but their contents/profile).
Response: According to the reviewer, this title was rewritten as follows: Volatile profile.
Point 13: As previously mentioned – tables and figures should be provided in the manuscript.
Response: Tables and figures have been provided in the version made by the journal, so they are included in the revised version.
ADDITIONAL COMMENTS FOR TABLES AND FIGURES:
Point 14: Table 1: Title should reflect more clearly what’s in the table
Response: According to the reviewer, the title of Table 1 has been modified as follows: Table 1. Agronomic and physicochemical parameters of Picual Organic Extra Virgin Olive Oil (OEVOO) samples depending on the sampling day (1, 15 and 32) and the time of collection (morning or afternoon) and the washing process.
Point 15: I would avoid using text in capital letters in the rows’ headings
Response: According to the reviewer, the text in rows was not included in capital letters.
Point 16: Explanation of “TT” should be revised (ANOVA and Tukey’s test are two different things and statement “analysis of variance by tukey's test” is incorrect); Explanation of the meaning of lowercase and capital letters seems to be incorrect and should be carefully revised (first of all it’s about difference between values within the same columns and not the same rows, and secondly there are e.g. no washing parameters in the table). Moreover, letters showing significance (or lack of significance) of differences between numbers should be placed next to every number and not in a separate row (what if there was a significant difference? How would this be indicated in this table?). Finally, why is Table 1 placed in the methodological section, and not in the Results section?
Response: As the reviewer said, ANOVA and Tukey is not the same, it was a mistake in the Table statement, but in materials and methods was correctly expressed. However, as no significant differences were observed, the rows with lowercase and capital letters have been deleted, and a new sentence has been included in the text explaining this as follows: “All the results are shown in Table 1. The Tukey test results did not show any significant difference between the parameters according to the sampling day, the time collection (i.e., morning or afternoon) or the washed and non-washed samples.” Moreover, Table 1 has been moved to results section and a Note has been included indicating the following: Note: Statistical evaluation results according to Tukey test were not included due to any significant difference was found.
Point 17: Figure 1, Figure 4, Figure 5: In the titles there is no information that it’s about olive oil. Figure title should be revised to reflect clearly what’s on the figure.
Response: According to the reviewer, the figure captions have been revised and changes have been made in order to clarify each figure. Moreover, Figure 4 and 5 have been fused in one due to they were related to the same PCA model, only changing the combination of PCs plotted.
Point 18: Table 2: Again – the title should reflect clearly what’s in the table (Sensory profile of Picual Organic Extra Virgin Olive Oil depending on…); in the footnote: A – Afternoon (and not “Afertnoon”); It should be specified in the table whether these are taste attributes (or odour); Statistical evaluation results are missing in the table; I would also suggest to specify (e.g. in the footnote) what the scale was for evaluation of these attributes.
Response: According to the reviewer’s suggestions, the title of Table 2 has been rewritten as follows: Sensory profile of Picual Organic Extra Virgin Olive Oil (OEVOO) depending on the sampling day (1, 15 and 32) and the time of collection (morning or afternoon) and the washing process. Moreover, “afternoon” has been corrected, and the specification that they are odour attributes and the specific scale are included in the footnote of Table 2.
Point 19: Figure 2: Again, it should be specified if the attributes in the figure = taste attributes; Each figure should be self-explanatory, thus I would suggest not to use abbreviation in the title (OEVOOs), but full name; The meaning of stars shown on the figure should be explained in the footnote.
Response: the attributes were odour attributes as it was indicated in Materials and methods. In order to clarify it again, it was included in Figure 2 caption. Furthermore, according to the reviewer, we have not used only the abbreviations in the figure captions or table titles. The explanation of the stars has been also included in the footnote of the figure.
Point 20: Figure 3: It should be clarified what the Authors mean by “total areas”; each Figure should be self-explanatory, thus I would suggest not to use abbreviation in the title (OEVOOs), but full name; Label should be added to y axis.
Response: The “total areas” has been changed to “sum of areas” in the title and in the text in order to clarify this aspect. OEVOOs has been changed to the full name and label of y axis has been included in the revised version.
‘TECHNICAL’ ASPECTS:
Point 21: Authors’ names should not be written in capital letters. Section and subsection titles – all words should begin with capital letters (except for “and”, “of” etc.).
Response: All these changes have been considered in the revised version of the manuscript.
Point 22: Current section 3 (Analytical determinations) should be a part of section 2 (Materials and Methods), and the following subsections of current section 3 should become subsections of section 2. Consequently, “Results” would become section 3 (and should be renamed as “Results and Discussion”), and “Conclusions” would become section 4.
Response: All these changes have been considered in the revised version of the manuscript.
“SUPPLEMENTARY MATERIALS” SECTION:
Point 23: The following text should be provided and the following formatting should be applied in this section of the manuscript: “The following supporting information can be downloaded at: www.mdpi.com/xxx/s1, Table S1: title; Table S2: title.” Moreover, it would be good if the Authors could add a specific heading to the Supplementary file, to make it clear that this is a Supplementary file related to this manuscript.
Numbering of tables in the Supplementary Materials should be done with Arabic numerals (S1, S2), and this should be revised accordingly in the whole manuscript text.
“Author Contributions” section: initials (and not full names of the authors) should be given.
“Data Availability Statement’ section should be added (after “Funding” section).
“Institutional Review Board Statement” section: Please revise (now the instructions from the manuscript template are written in this section…).
“Conflicts of Interest” section: add full stop at the end of the sentence.
Response: We are very grateful for all these suggestions. Therefore, all these changes have been considered in the revised version of the manuscript.
REFERENCES:
Point 24: Following the requirements of Foods (MDPI), all names of the journals in the listed references should be abbreviated. Other aspects should also be carefully revised, to ensure the style follows the requirements of the journal. At the moment there is a significant number of inconsistencies.
Response: We have fully revised the references in order to improve them according to the style of the journal.

Reviewer 2 Report
All the profiles of this manuscript is well presented. However, I have one suggestion on the definition of "harvesting time including morning and afternoon". How did the authors define the specific time, 8:00-12:00 am, as morning time? 12:00-18:00 pm, as afternoon time ??
Author Response
Response to Reviewer 2 Comments
Point 1: All the profiles of this manuscript is well presented. However, I have one suggestion on the definition of "harvesting time including morning and afternoon". How did the authors define the specific time, 8:00-12:00 am, as morning time? 12:00-18:00 pm, as afternoon time??
Response: The specific time of harvesting was the following: from 8:30 to 12: am as the morning time, and from 13:00 to 15:00 pm as afternoon time. This information has been included in the text, as suggested by the reviewer.
